# Survival Rates and Clinical Outcomes of Implant Overdentures in Old and Medically Compromised Patients

**DOI:** 10.3390/ijerph191811571

**Published:** 2022-09-14

**Authors:** So-Hyun Kim, Nam-Sik Oh, Hyo-Jung Kim

**Affiliations:** 1Department of Dentistry, Inha University Hospital, Inha University School of Medicine, 27 Inhang-ro, Jung-gu, Incheon 22332, Korea; 2Department of Dentistry, Ulsan University Hospital, University of Ulsan College of Medicine, 877 Bangeojinsunwhando-ro, Dong-gu, Ulsan 44033, Korea

**Keywords:** dental implants, systemic disease, implant overdenture, quality of life

## Abstract

Studies on the survival rate of implant overdentures in medically compromised patients are limited because most studies exclude patients with systemic diseases affecting implant prognosis. This retrospective study aimed to evaluate the survival rate and clinical outcomes of dental implants used for overdentures in medically compromised patients. A total of 20 patients (9 men, 11 women; mean age: 67.55 ± 6.84 years, range: 53–81 years) were included. Fourteen patients had more than two systemic diseases, and nine patients had more than three systemic diseases. The mean follow-up period was 39.05 months. Of the 60 implants, 2 failed, resulting in an implant survival rate of 96.6%. No statistical differences were found in implant survival rates according to sex, age, implant diameter, restored arch, or opposing dentition (*p* > 0.05). A significant difference in mean marginal bone loss (MBL) was noted for restoring the arch (*p* = 0.022) and opposing dentition (*p* = 0.036). Implants placed in the mandible and with opposing removable partial dentures and complete dentures showed lower mean MBL. No significant differences in implant MBL were observed in terms of age, sex, or implant diameter (*p* > 0.05). Favorable clinical outcomes can be expected from implant overdentures using two or four implants in edentulous patients with systemic diseases by ensuring that the patients have a sufficient healing period and regular checkups.

## 1. Introduction

Advances in medicine, a growing interest in health, and improvement in nutritional status are increasing the average life expectancy. In Korea, the average life expectancy is 80.5 years for men and 86.5 years for women. The proportion of the elderly population aged 65 and over in Korea was 15.7% in 2020 and is expected to continue increasing thereafter, exceeding 25.5% in 2030 and 46.4% in 2070 [1]. Based on the National Survey of Older Koreans over 65 years of age, participants had an average of 1.9 diseases: one disease in 29%, two in 27%, and three or more in 28%. The commonest diseases identified were high blood pressure (57%), diabetes (24%), hyperlipidemia (17%), osteoarthritis, and rheumatoid arthritis (17%) [2].

Since 2012, medical insurance has been applied to dentures in the elderly, and the number of dental visits by people with systemic diseases in the elderly population is rapidly increasing [3]. If the residual alveolar ridge is good or they do not feel much discomfort in using their dentures, it may be sufficient to make new dentures. However, if absorption of the residual alveolar ridge is profound due to the use of ill-fitted dentures for a long time or, in the case of dry mouth, due to polypharmacy, it is difficult to adapt to dentures because of the underlying tissue pain under the denture and poor denture retention. Chronic dry mouth occurs in a significant proportion of older adults and affects oral-health-related quality of life, including speaking, enjoying and eating food, and wearing prosthetics [4]. 

Implant overdentures (IOD) improve support under the denture base to relieve pain and improve retention and stability [5,6], and it is possible to improve aesthetic appearance through soft tissue support that fixed prostheses cannot provide. In addition, there is an advantage that a family member or caregiver can easily manage oral hygiene if the patient cannot detach it by himself or herself [7]. Currently, an IOD with two implants, as suggested by the McGill consensus in 2002, is considered the standard treatment option for patients with complete edentulous mandibles [8]. A literature review suggested that mandibular 2-IOD is the minimum standard that should be sufficient for most people, taking into account performance, patient satisfaction, cost, and clinical time [9]. For edentulous mandibles, IODs with two or four implants have been shown to improve oral-health-related quality-of-life outcomes and have proven to be cost-effective [10]. 

Generally, systemic diseases and old age are considered high-risk factors for dental implant surgeries. Although dental implants are used in medically compromised patients, it is often not well-known whether implant treatment is adequate in these patients, whether the risk of implant failure and peri-implantitis is increased, and what preventive measures should be taken when applying dental implants in this patient group [11]. Although dental implant surgery is not absolutely contraindicated in edentulous patients with systemic disease, most are treated with complete dentures (CDs). Considering the patient’s age and systemic disease(s), fixed implant prostheses require six to nine implants [12], which are expensive and unsuitable for patients who cannot independently manage oral hygiene. On the other hand, an IOD, which involves the placement of a small number of implants, is cost-effective and easy to adapt because it is familiar to patients who have used dentures in the past. Wolfart et al. reported that patients in an edentulous group benefited from strategically placed implants under the existing dentures, and chewing satisfaction generally increased after implant therapy [6]. 

Studies on the survival rate of IODs in medically compromised patients are limited because in most studies, patients with systemic diseases (e.g., uncontrolled diabetes) affecting implant prognosis have been excluded. This study aimed to evaluate the survival rate and clinical outcomes of dental implants used for overdentures in medically compromised patients. 

## 2. Materials and Methods

Among patients who were treated with IODs at Ulsan University Hospital in South Korea between June 2015 and June 2021, this study focused on those with systemic diseases. This study was approved by the institutional review board of Ulsan University Hospital (IRB protocol no: 2021-07-016-003). Patient consent was waived due to the retrospective nature of this study. 

Sex, age, systemic disease, implant width, restored arch, opposing dentition, marginal bone loss (MBL), and prosthetic complications were investigated by reviewing the patients’ medical records and radiographs. A total of 20 patients were included: 9 men and 11 women. Patient ages ranged from 53 to 81 years (mean age: 67.55 ± 6.84 years). Fourteen patients had more than two systemic diseases, and nine patients had more than three systemic diseases. Diabetes mellitus (DM) was the commonest single disease and reported in 13 patients, followed by hypertension (HTN) in 10, cancer in 5, and osteoporosis in 5. The baseline characteristics of the patients and implants are presented in Table 1. 

Four implants were placed in the edentulous maxilla, and two to four implants were placed in the mandible. A total of 60 bone-level internal implants were placed using the submerged method. After a healing period of 3–6 months (mandible) and 6–11 months (maxilla), second-stage surgery was performed. The locator, magnet, or customized bar attachments were connected to the implants, and the female part of the attachments was integrated into the denture. All IODs were manufactured by a prosthodontist. Patients were instructed to maintain oral hygiene control and visit the hospital every 6 months for regular checkups and were encouraged to contact the hospital if any problems occurred. 

The implants were divided into a less than 3.75 mm width group and a more than 3.75 mm width group. Implants with a diameter of less than 3.75 mm were called narrow-diameter implants (NDIs), and those with a diameter of more than 3.75 mm were called regular-diameter implants. In addition, the implants were classified according to the dental arch in which they were placed. The implants were categorized according to opposing dentition. A panoramic radiograph was obtained at implant installation, after implant loading, and at regular follow-up visits. Marginal bone loss (MBL) was calculated as the difference in the bone level between IOD delivery and the final follow-up visit. Prosthetic complications were also recorded. The implant survival rates were also determined. The implant survival criteria followed the recommendations of the 2007 Pisa consensus. The implant was considered to survive if the implant and its superstructure remained normally functioning at the point of the final observation. Implant failure was defined as implants that required removal or were already lost. The implant was indicated for removal under any of the following conditions: (1) pain on palpation, percussion, or function; (2) horizontal and/or vertical mobility; (3) uncontrolled progressive bone loss; (4) uncontrolled exudate; or (5) > 50% bone loss around the implant. Implants that were surgically placed but unable to be restored (sleepers) were also included in failure determination [13]. Prosthesis survival was defined as a condition that can be repaired by minor complications, such as denture base fracture, artificial tooth fracture, and relining. Failure of dentures was defined as a condition that required remaking owing to the fracture of the denture framework. 

All statistical analyses were performed using IBM SPSS Statistics version 24 (IBM Corp., Armonk, NY, USA). The Kaplan–Meier analysis method was used for implant survival analysis with the log-rank test used to compare variables. The Mann–Whitney U and Kruskal–Wallis tests were used for MBL analyses. Statistical significance was set at *p* value < 0.05.

## 3. Results

Sixty implants of 23 IODs in 20 patients were evaluated (three patients were treated with two IODs in both the maxilla and mandible). Table 1 shows the baseline characteristics of our cohort. Of the 23 IODs, five were placed on the maxilla and 18 on the mandible. The mean follow-up period was 39.05 ± 16.05 months (minimum, 16 months; maximum, 69 months). No IOD remake was performed, yielding a 100.0% prosthesis survival rate. During the study period, two of the 60 implants failed. This resulted in an implant survival rate of 96.6%. 

Among the 20 patients, 34 implants were placed in nine male patients and 26 implants were placed in 11 female patients. Two implants failed in both male and female patients, resulting in implant survival rates of 97.1% and 96.2%, respectively. There was no significant difference in the survival rates according to sex (*p* = 0.456; Table 2). The Kaplan–Meier survival curve according to sex is shown in Figure 1.

Patient ages ranged from 53 to 81 years (mean age: 67.55 ± 6.84 years). According to the age distribution, one patient was 81 years old, six patients were 70–79 years old, 11 patients were 60–69 years old, and two patients were under 60 years old. Based on the age threshold of 65 years, 26 implants were inserted in patients under 65 years of age (six patients), and 34 implants were inserted in patients over 65 years of age (14 patients). The survival rates of implants in patients under 65 and over 65 years of age were 96.2% and 97.1%, respectively. No significant difference was noted in the survival rates (*p* = 0.786; Table 3). The Kaplan–Meier survival curve according to age is shown in Figure 2.

Of the 60 implants placed, nine implants were of 4.5 mm width, 37 implants were of 4.0 mm width, and 14 implants were of less than 3.75 mm width. Of those less than 3.75 mm in width, two failed in two patients. None of the implants more than 3.75 mm in width failed. The survival rates of the narrow and regular implants were 85.7% and 100.0%, respectively. There was no significant difference in the survival rates between narrow- and regular-diameter implants (*p* = 0.099; Table 4). The Kaplan–Meier survival curve according to the implant diameter is shown in Figure 3.

Twenty implants were placed in the maxilla and 40 in the mandible. In the maxilla, one implant failed before functional loading, resulting in a survival rate of 95.0%. In the mandible, one implant failed within 25 months of functional loading, resulting in an implant survival rate of 97.5%. No significant difference was noted in the survival rates (*p* = 0.919; Table 5). The Kaplan–Meier survival curve according to the restored arch is illustrated in Figure 4.

Of the eight implants with opposing natural teeth and implant-supported fixed prostheses (ND), one failed, resulting in an 87.5% survival rate. Of the 20 implants with opposing IODs, one implant failed, resulting in a 95.0% survival rate. Of the six implants with opposing removable partial dentures (RPDs) and 26 implants with opposing CDs, there was no implant failure among the 32 implants, which resulted in a 100.0% survival rate. There was no significant difference in survival rates according to the opposing dentition (*p* = 0.263; Table 6). The Kaplan–Meier survival curve according to the opposing dentition is shown in Figure 5.

Table 7 shows the analysis of the MBL around the implants, excluding the two failed implants. The MBL of the implants showed significant differences in the restored arch (*p* = 0.022) and opposing dentition (*p* = 0.036). The mean MBL of the maxilla and mandible were 0.53 ± 1.34 mm and 0.06 ± 0.39 mm, respectively. There was significantly higher mean MBL in the maxilla than in the mandible (*p* = 0.022; Table 7). The mean MBL of the ND, IOD, and RPD and CD groups were 0.08 ± 0.22 mm, 0.63 ± 1.41 mm, and 0.00 ± 0.00 mm, respectively. A significant difference in the mean MBL between the types of opposing dentition was observed (*p* = 0.036; Table 7). The implants with opposing RPD and CD showed lower mean MBL than ND or IOD. No significant difference in the implant MBL was observed in terms of age (*p* = 0.076), sex (*p* = 0.263), or implant diameter (*p* = 0.202). Three of the 58 implants showed more than 2.0 mm bone loss in one patient with DM. The patient wore his dentures all day and had poor oral hygiene.

According to the attachment type, 40 implants were connected to the locator attachment, 15 implants had customized bar attachments, and the remaining four implants were attached to the magnet. As for prosthetic complications, the most frequent complication was dislodgement of the locator nylon matrix and hader clip. Two cases of wear and chipping of the artificial teeth occurred, and two cases of locator wear occurred when the long axis of each implant was not parallel. No locator replacement was performed during the study period. 

## 4. Discussion

This retrospective study included 20 patients with systemic diseases. During the study period, two of the 60 implants failed, resulting in an implant survival rate of 96.6%. No statistical differences were found in implant survival rates according to sex, age, implant diameter, restored arch, or opposing dentition (*p* > 0.05). The MBL of the implants showed significant differences in relation to the restored arch (*p* = 0.022) and opposing dentition (*p* = 0.036) involved. Of the two failed implants, one failed to osseointegrate and was sleeping without functional loading. Patient #16 had been using the IOD supported by a customized bar connected to the three remaining implants placed in the anterior region of the maxilla without any problems (follow-up period, 44 months). The opposing dentitions were the anterior natural teeth and posterior implant-supported fixed restorations. He had DM, HTN, and asthma. The other failed implant was one of the two implants connected to the locator attachments in the anterior mandible. It was removed because of peri-implantitis within 25 months after functional loading. The lost implant was subsequently replaced. Wear of the locator attachment was observed because the axes of the two implants were not parallel, and the artificial resin posterior teeth of the opposing IOD were fractured before implant failure. Patient #18 said that she wore dentures all day, including when sleeping, and had DM and osteoporosis. Two of the failed implants were in DM patients, and 13 of the 20 patients in this study had DM. Most other studies on implant survival have included patients with well-controlled DM; however, this study included two patients with uncontrolled DM. Furthermore, 14 patients had more than two systemic diseases, and nine patients had more than three systemic diseases. According to Dudley, although individual relative contraindications may not rule out implant treatment, combinations of relative contraindications may be collectively equivalent to absolute contraindications [14]. With multiple chronic conditions, their effect on implant treatment is complex and poorly understood [15]. 

DM is the most prevalent endocrine disease [16] and is associated with delayed wound healing and impaired bone healing. DM may influence both the short-term prognosis of implant therapy, which depends on successful osseointegration, and the long-term prognosis, in which bone remodeling plays a role in responding to the functional demands after implant loading. As a result, the altered healing response in DM patients results in less bone formation and less bone–implant contact, which could make the implants less resistant to micromovement and more prone to failure. To minimize the risk of implant failure, systemic reviews and clinical studies recommend strict metabolic control before, during, and after implantation; antibiotic prophylaxis; chlorhexidine digluconate rinsing; experienced surgeons; and short recall intervals [17,18]. Patients with poorly controlled diabetes have been shown to have an increased risk of developing peri-implantitis [11,19]. De Oliveira et al. found that the risk of peri-implantitis increases with hyperglycemic conditions in an independent manner. A proportional relationship between peri-implant crestal bone maintenance and glycemic levels has been observed; thus, the group with higher hemoglobin A1C (HbA1c) levels had greater MBL than controls [20]. The results of a systematic review and meta-analysis identified that HbA1c levels above 8% may result in reduced implant survival compared with lower levels [15]. However, well-controlled diabetes does not impose any additional risk for patients undergoing dental implant therapy [18]. Sghaireen et al. reported that out of 377 dental implants placed in diabetic patients, 17 (4.50%) failed after the first stage of surgery, indicating that they were not properly osseointegrated. The overall survival rate after 3 years of follow-up in the diabetic and nondiabetic patients was 90.18% and 90.95%, respectively. They suggested that an antiplaque agent (chlorhexidine 0.12%) and regular maintenance were efficient in decreasing the implant failure rate in patients with diabetes [18]. Thirteen (65%) of the 20 patients in this study had DM, including two patients with uncontrolled DM (HbA1c ≥ 8.5). Uncontrolled DM was an exclusion criterion in most of the other studies. Interestingly, the six implants placed showed no significant bone loss and did not fail. However, three implants placed in one patient with DM had significant crestal bone loss at the time of prosthesis connection, and additionally, more than 2.0 mm of bone loss occurred. The patient wore his dentures all day and had poor oral hygiene. 

Granato et al. showed that, although implant survival rates were similar between healthy and metabolic syndrome (MS) patients, MS significantly reduced bone formation in the peri-implant area in the short term. They suggested that, given the increasing prevalence of MS patients requiring implant treatment, the bone response to implants should be considered when determining the ideal loading time in this population [21]. In the present study, all implants were placed using a two-stage approach, and in most cases, a longer healing period was secured than the conventional healing period of 6 months for the maxilla and 3 months for the mandible. According to Kern et al., both immediate and conventional loading protocols exhibited low implant loss rates, and no statistically significant differences were observed in fixed restorations in both jaws. However, there is a significantly lower risk of implant loss with a conventional loading protocol concerning the overall analysis, removable prostheses, and edentulous mandibles [12]. 

Five patients with osteoporosis were included in this study. One of the 14 implants failed. Patients with bone metastases, including breast and prostate cancer or those with multiple myeloma, often receive high-dose intravenous antiresorptive therapy (ART), which may be associated with medication-related osteonecrosis of the jaw (MRONJ) [15]. It has been reported that the anterior mandibular region does not experience significant age-related osteopenia when considering implant treatment for elderly patients [14]. In studies of osteoporotic patients managed with ART, the reported implant survival rates are largely high, and the prevalence of MRONJ in these patient cohorts is rarely specified [11,15]. As bone density in osteoporotic patients is lower, a longer healing time is recommended before functional loading [11]. 

Patients with neurological diseases such as dementia and Parkinson’s disease are excluded from implant treatment. The main reasons for exclusion are poor access to oral healthcare, poor oral hygiene, and oral parafunctions. Treatment of edentulism with CDs may present a challenge for patients with neurological diseases, and edentulism can lead to a deficient nutritional intake. There is little evidence to support the use of dental implants in patients with neurologic diseases [19]. In the present study, two patients with dementia, one with ataxic cerebral palsy, and one with epilepsy were included. Of the 10 implants, none failed, and the MBL was no more than 1 mm. No specific prosthetic complications were found other than retention loss due to the wear of the locator nylon matrix. We assumed that the reason for the good outcome was that the patients were undergoing regular checkups, and the opposing dentitions were maxillary removable prostheses. Manor et al. reported similar rates of complications and implant failures in medically compromised and healthy patients and suggested the importance of performing implant surgery with strict asepsis, minimal trauma, avoidance of stress in patients with systemic disease, and maintenance of optimal oral hygiene and smoking cessation [22]. 

Of the 60 implants in our study, two NDIs failed. None of the regular-diameter implants failed. NDIs have the advantage of reducing the need for invasive surgery in older patients or patients with surgical risk factors. A meta-analysis was conducted on NDIs (3.3–3.5 mm), which showed no statistically significant difference in implant survival compared with regular-diameter implants [23]. In another study, as the implant diameter increased, the stress and strain at the implant–bone interface significantly decreased, especially when the diameter increased from 3.3 mm to 4.1 mm [24]. The results of a three-dimensional finite-element analysis showed that stress values and concentration areas decreased in the cortical bone when the implant diameter increased [25]. 

There is no doubt that implants are now one of the most common treatments in dentistry. In general, elderly patients have various systemic diseases, and it is assumed that their bone healing ability and immune function are reduced [26]. The demand for dental implant treatment for the elderly and patients with systemic diseases is gradually increasing, and it is expected to increase further owing to the increase in the elderly population [1]. However, many dentists tend to exclude patients with systemic diseases from dental implants to reduce the risk and provide stable treatment. The Korean National Health Insurance Service (NHIS) provides insurance for oral reconstruction procedures, including complete dentures, removable partial dentures, and dental implants. Seo et al. reported that the Korean dental health insurance policy has been beneficial for the medical expenses of low-income and elderly individuals suffering from a cost burden due to systemic diseases. However, since there is a tendency to avoid invasive interventions in older patients due to the high risk of systemic diseases, they suggested that insurance coverage of dentures may be more helpful from a socioeconomic perspective than coverage of dental implant treatments [3]. Systematic reviews and clinical studies have recommended that dental implant treatment is accompanied by significant functional benefits and improved oral-health-related quality of life. Dental implant therapy is an adequate treatment in almost any medically compromised patient when the required preventive measures are taken and follow-up care is at a high level [11,15,22,27]. Clinical decision-making should consider the oral and systemic health of patients with comorbidities in the form of an individualized risk assessment that includes close collaboration with healthcare professionals and family physicians [15]. According to Diz et al., the degree of systemic disease control may be far more important than the nature of the disorder itself, and individualized medical control should be established prior to implant therapy because in medically compromised patients, the quality of life and functional benefits from dental implants may outweigh any risks [27]. 

Nowadays, the demand for quality of life is emphasized more than ever, and oral health is directly linked to overall health. Oral mastication is the first gateway to the digestive system and contributes to the prevention of systemic frailty by enabling the ingestion of a variety of foods. Ramsay et al. showed that oral health problems were associated with greater risks of being frail and developing frailty in the elderly, and management of poor oral health in older people could be important in preventing frailty [28]. Appollonio et al. reported that denture wearers had a dietary intake very similar to those with adequate dentition and were substantially better than those with inadequate dentition [29]. Bakker et al. showed that the general health of the elderly who received an IOD was comparable with that of the elderly with a natural dentition and better than that of the elderly with a CD [30]. Recent studies have reported that oral dysfunction of the masticatory system is closely related to cognitive decline in the elderly [31,32]. IODs are relatively easy to adapt to and have a high masticatory efficiency because support and retention can be obtained by placing a small number of implants under the dentures [6]. Marotti et al. suggested that the therapeutic concept of implant placement under existing prostheses is promising when performed in indicated cases [33]. Dependent elderly patients with mandibular dentures can benefit from the insertion of dental implants, providing adequate oral healthcare and aftercare by caregivers. Regular information and instructions for caregivers and family members regarding the oral condition of the patient are essential for proper maintenance [7]. A prospective study to evaluate whether age affects peri-implant health in patients treated with mandibular 2-IOD for 10 years reported that implant survival rates were 97.1% and 93.4% in the younger (n = 52; mean age 45 years) and older groups (n = 53; mean age 68 years), respectively. No significant differences were observed between the groups [34]. In a prospective study with a 20-year follow-up, Bakker et al. showed that eight out of 106 implants were lost, resulting in a 92.5% implant survival rate of mandibular IODs in (frail) elderly individuals (aged ≥60 years), and participants were very satisfied with their prosthesis and reported a good quality of life. Despite the frailty and deteriorated oral hygiene of the participants, their results showed that the IOD is a durable treatment option. They speculated that continuing to visit the dentist on a yearly basis might be an important factor in preventing severe peri-implantitis [5]. 

Prosthetic complications include retention loss due to wear of the locator nylon matrix or hader clip, and artificial teeth wear and chipping. In one case, the male part of the locator attachment was damaged due to nylon matrix dropout when a patient suddenly came to the hospital after a long time without attending regular checkups. Regular checkups are important for the prognosis of IODs. Patients should be advised of their role in maintenance, and a comprehensive recall system is mandatory to obtain satisfactory long-term results [35]. Other studies have also reported that retention loss of the attachment system is the most frequent complication [33,36]. 

Dental prostheses should be easy to clean, particularly for the elderly. As oral hygiene may be compromised in older patients with cognitive or motor impairment, proper and regular instructions should be given not only to patients but also to relatives or caregivers [37]. In a two-session within-subject crossover trial comparing maxillary implant-retained fixed prostheses with IODs, patients rated their ability to speak and ease of cleaning significantly better with IODs [38]. The choice of the attachment system is important in patients who may have difficulty maintaining oral hygiene. Unsplinted implant attachments (e.g., locators) may offer greater ease of hygiene and maintenance or repair than splinted attachment systems [14]. A systematic review reported the superiority of IODs retained by two unsplinted mandibular implants compared with CDs in terms of efficacy, satisfaction, and quality of life [39]. Stoumpis et al. reported that after 3 years, there was no difference in implant survival between splinted and unsplinted implants [40]. The maxillary overdenture implant survival rate is the lowest among all implant prosthetic types and has been reported to be as low as 71% over 5 years [14]. Anadioti et al. showed that unsplinted maxillary IODs were associated with high implant and prosthetic survival as well as high patient satisfaction and quality of life [41]. In our study, according to the attachment type, 44 implants were connected to solitary attachments (40 implant locators and four implant magnet attachments). 

In our study, 60 bone-level internal implants were placed using the submerged method. A recent systemic review showed no differences in the MBL or implant survival rate between bone-level and transmucosal dental implants after a period of follow-up ranging from 12 to 60 months [42].

The present clinical study included only patients with systemic diseases and did not compare the outcomes with those of healthy controls. Although the observation period was relatively short (39.05 ± 16.05 months), these findings are comparable with other studies in healthy patients. The current findings are expected to contribute to improving the quality of life of patients excluded from implant treatment due to multiple systemic diseases. There were some limitations in the present study, including the limited number of subjects and the short observation period. Future research designs should be standardized, and prospective clinical research should be conducted to produce reliable and generalizable results.

## 5. Conclusions

Within the limitations of the present study, favorable clinical outcomes can be expected from implant overdentures using two or four implants in edentulous patients with systemic diseases by ensuring that the patients have a sufficient healing period and regular checkups. It will be possible to provide medically compromised patients who are excluded from implant treatment owing to multiple systemic diseases with a better quality of life by considering information on systemic diseases and medications. Further long-term studies are required to confirm the results of the present study. Additional results will be reported in the future when long-term follow-up is performed.

## Figures and Tables

**Figure 1 ijerph-19-11571-f001:**
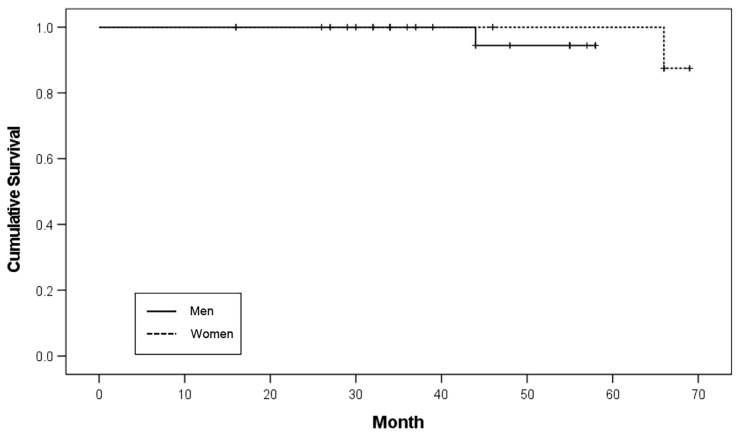
Kaplan-Meier survival curve according to sex.

**Figure 2 ijerph-19-11571-f002:**
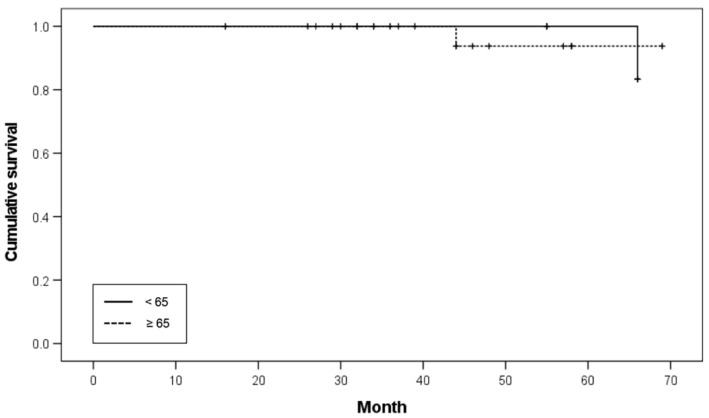
Kaplan-Meier survival curve according to age.

**Figure 3 ijerph-19-11571-f003:**
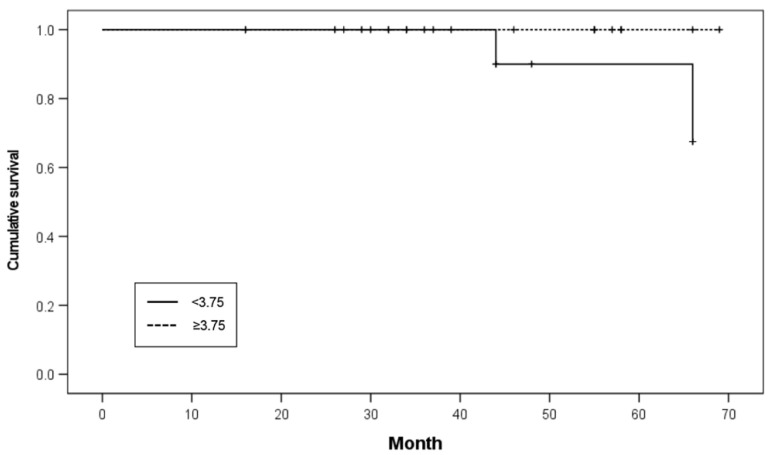
Kaplan–Meier survival curve according to implant diameter.

**Figure 4 ijerph-19-11571-f004:**
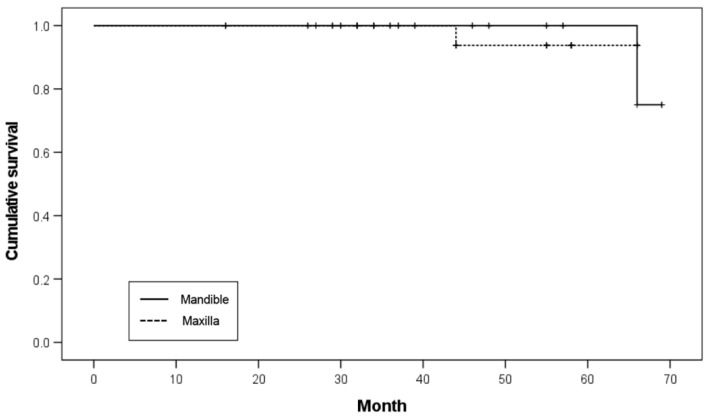
Kaplan–Meier survival curve according to restored arch.

**Figure 5 ijerph-19-11571-f005:**
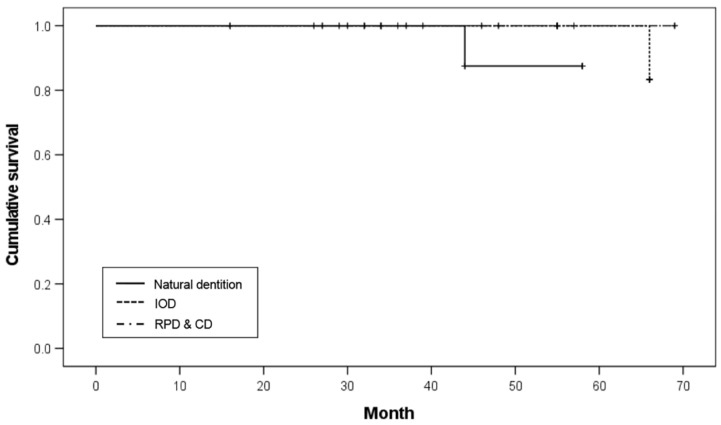
Kaplan–Meier survival curve according to opposing dentition.

**Table 1 ijerph-19-11571-t001:** Baseline characteristics of patients and implants.

Patient	Sex	Age	Systemic Disease	Restored Arch	Opposing Dentition	Number of Implants	Follow-Up Period (Month)
1	W	72	DM, HTN, O	Mn	CD	2	69
2	W	78	DM, HTN, dementia	Mn	RPD	2	26
3	M	74	DM	Mn	RPD	2	57
4	W	81	O, depression	Mn	RPD	2	46
5	M	75	DM, HTN, CA (stomach)	Mn	CD	2	48
6	W	66	DM	Mn	CD	2	39
7	W	66	DM, CA (thyroid), CI	Mn	CD	2	37
8	M	63	DM	Mx/Mn	IOD	4/4	34
9	W	64	HTN, dementia	Mn	CD	2	29
10	M	68	DM, HTN, CI	Mn	CD	2	16
11	M	72	DM (uncontrolled), ALC	Mn	CD	2	27
12	M	67	HTN, CA (oral, colon, lung)	Mx	ND	4	58
13	M	53	DM (uncontrolled)	Mx/Mn	IOD	4/2	55
14	M	67	DM, HTN, epilepsy	Mn	CD	4	32
15	W	67	HTN, CA (ovary, colon), O	Mn	CD	2	30
16	M	68	DM, HTN, asthma	Mx	ND	4	44
17	W	57	HTN, O, cerebral palsy	Mn	CD	2	36
18	W	60	DM, O	Mx/Mn	IOD	4/2	66
19	W	71	O	Mn	CD	2	16
20	W	62	CA (liver, ovary)	Mn	CD	2	16

ALC, alcoholic liver cirrhosis; CA, cancer; CD, complete denture; CI, cerebral infarction; DM, diabetes mellitus; W, women; HTN, hypertension; IOD, implant overdenture; M, men; Mx, maxilla; Mn, mandible; ND: natural dentition or implant-supported fixed prostheses; O, osteoporosis; RPD, removable partial dentures.

**Table 2 ijerph-19-11571-t002:** Survival rate according to sex.

Sex	Placed Implants (n)	Failed Implants (n)	Survival Rate (%)	*p* Value
Men	34	1	97.1	0.456
Women	26	1	96.2	

**Table 3 ijerph-19-11571-t003:** Survival rate according to age.

Age (y)	Placed Implants (n)	Failed Implants (n)	Survival Rate (%)	*p* Value
<65	26	1	96.2	0.786
≥65	34	1	97.1	

**Table 4 ijerph-19-11571-t004:** Survival rate according to implant diameter.

Width	Placed Implants (n)	Failed Implants (n)	Survival Rate (%)	*p* Value
Narrow (<3.75 mm)	14	2	85.7	0.099
Regular (≥3.75 mm)	46	0	100.0	

**Table 5 ijerph-19-11571-t005:** Survival rate according to restored arch.

Restored Arch	Placed Implants (n)	Failed Implants (n)	Survival Rate (%)	*p* Value
Maxilla	20	1	95.0	0.919
Mandible	40	1	97.5	

**Table 6 ijerph-19-11571-t006:** Survival rate according to opposing dentition.

Opposing Dentition	Placed Implants (n)	Failed Implants (n)	Survival Rate (%)	*p* Value
ND	8	1	87.5	0.263
IOD	20	1	95.0	
RPD and CD	32	0	100.0	

CD, complete denture; IOD, implant overdenture; ND, natural dentition or implant-supported fixed prostheses; RPD, removable partial denture.

**Table 7 ijerph-19-11571-t007:** Marginal bone loss of implants (mm).

		Implant (n)	Bone Loss (Mean ± SD)	*p* Value
Sex †	Men	34	0.35 ± 1.11	0.263
	Women	26	0.05 ± 0.26	
Age †	<65	26	0.48 ± 1.26	0.076
	≥65	34	0.02 ± 0.10	
Implant diameter †	Narrow (<3.75 mm)	14	0.00 ± 0.00	0.202
	Regular (≥3.75 mm)	46	0.29 ± 0.97	
Restored arch †	Maxilla	20	0.53 ± 1.34	0.022
	Mandible	40	0.06 ± 0.39	
Opposing dentition ‡	ND	8	0.08 ± 0.22	0.036
	IOD	20	0.63 ± 1.41	
	RPD and CD	32	0.00 ± 0.00	

† Mann–Whitney U test, ‡ Kruskal–Wallis test. CD, complete denture; IOD, implant overdenture; ND, natural dentition or implant-supported fixed prostheses; RPD, removable partial denture; SD, standard deviation.

## Data Availability

Not applicable.

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
