# Peer review of "Survival Rates and Clinical Outcomes of Implant Overdentures in Old and Medically Compromised Patients"

_ijerph, 2022, doi:10.3390/ijerph191811571_

Round 1
Reviewer 1 Report
It would be interesting to have a control group, that is, patients with the same characteristics (gender, age, number of implants, type of prosthesis) to be able to compare with the group with systemic diseases. A weak point in the work is the follow-up interval of approximately 3 years and also the radiographic evaluation is performed by means of panoramic radiography, which presents distortions in the anterior region (location of the implants in this study). But this does not detract from the authors' merit, on the contrary, it emphasizes the social importance of studies like this one. Congratulations.
Author Response
Please see the attachment.
Ms word file contains responses to reviewers 1,2,3.

Reviewer 2 Report
Thank you for your research and submission regarding the paper “Survival rates and clinical outcomes of implant overdentures in old and medically compromised patients”.
Samples with 60 implants (20 patients) might seem small but the importance of the study relies on the realistic situation of the age group used: elderly usually have some illnesses and most papers use samples of non-medically compromised mature adults.
Although the manuscript is quite clear, some little details need to be corrected before continuing.
Abstract: the significant differences described in lines 20-21 need to be explained, that is, which groups present significant higher or lower MBL.
Introduction: the section is well developed, with interesting references. But results should not be presented, therefore the phrase regard “Our results…” after the objective, at the end of the section should be removed. Please note that you already reproduce it in the conclusion section…
Material and Methods section: why dividing the width using a cut-off value of 3.75 mm? (provide some explanation in the text).
Statistical analysis is well explained and performed.
Results section: I’m used to see the legend of graphs after they are present (the legend of the tables is correctly presented before they appear).
On table 4 you should remove the underline under higher or equal sign) and include de units (mm) after the 3.75.
On table 7, a column with group size (n) should be included, as it is easier to understand if the non‑significance is also due to a difference in group size. The same “critic” regarding including de units (mm) after the 3.75 applies here. In the text, line 207-208, explain the significant difference, that is, explain that a significantly higher marginal bone loss was found for the Maxilla implants than for the mandible. Regarding the comparison of the three groups with Kruskal-Wallis (table 7) and after obtaining a significant difference we would benefit from knowing if all three groups differ in MBL of if only IOD differs (being the MBL significant higher in IOD) from the two other groups (ND and RPD&CD) – for that you should present (in the text) the P-values for the pair-wise comparison (using the Mann-Whitney test).
Discussion and conclusion might have some rearrangements but overall are well done.
Reviewer 3 Report
Dear Author, the paper is very interesting and the topic is very hot in Dentistry right now but the sample is very short and I suggest you to call this article "a retrospective pilot study" hoping to do more in future analysis.
In fact you should state the numerous limitations of the study in discussion citing studies that have. focused on the topic. Please mention that you do not have considered the fact that the implants may be bone-level or tissue level, because after some years if there is no platform switching over the bone-level implants may have more marginal bone loss compared with tissue level one. You must cite the last systematic review of this argument:
Cosola S, Marconcini S, Boccuzzi M et all. Radiological outcomes of bone-level and tissue-level dental implants: Systematic review. 2020
You should state in the materials and method the type of implants and the prosthetic components. The knowledge of systematic disease is also not clear. Since the moment of the diagnosis how much time passed with dental implants? Please improve materials, results and discussion
Best regards
